# Chitinase-3-like Protein 1 Is Associated with Poor Virologic Control and Immune Activation in Children Living with HIV

**DOI:** 10.3390/v14122602

**Published:** 2022-11-23

**Authors:** Isabelle Bernard, Doris G. Ransy, Jason Brophy, Fatima Kakkar, Ari Bitnun, Lindy Samson, Stanley Read, Hugo Soudeyns, Michael T. Hawkes

**Affiliations:** 1Department of Pediatrics, University of Alberta, Edmonton, AB T6G 2R3, Canada; 2Unité d’immunopathologie Virale, Centre de Recherche du CHU Sainte-Justine, Montreal, QC H3T 1C5, Canada; 3Division of Infectious Diseases, Children’s Hospital of Eastern Ontario, Ottawa, ON K1H 8L1, Canada; 4Department of Pediatrics, University of Ottawa, Ottawa, ON K1N 6N5, Canada; 5Department of Pediatrics, CHU Sainte-Justine, Université de Montréal, Montreal, QC H3T 1C5, Canada; 6Division of Infectious Diseases, Hospital for Sick Children, Department of Pediatrics, University of Toronto, Toronto, ON M5G 1X8, Canada; 7Department of Microbiology, Infectiology & Immunology, Université de Montréal, Montreal, QC H3T 1J4, Canada; 8School of Public Health, University of Alberta, Edmonton, AB T6G 1C9, Canada; 9Department of Medical Microbiology and Immunology, University of Alberta, Edmonton, AB T6G 1C9, Canada; 10Stollery Science Lab, Edmonton, AB T6G 1C9, Canada; 11Women and Children’s Health Research Institute, Edmonton, AB T6G 1C9, Canada

**Keywords:** child, HIV, chitinase-3-like protein 1, inflammation, neutrophil, microbial translocation

## Abstract

Perinatally infected children living with HIV (CLWH) face lifelong infection and associated inflammatory injury. Chitinase-like 3 protein-1 (CHI3L1) is expressed by activated neutrophils and may be a clinically informative marker of systemic inflammation in CLWH. We conducted a multi-centre, cross-sectional study of CLWH, enrolled in the Early Pediatric Initiation Canadian Child Cure Cohort Study (EPIC^4^). Plasma levels of CHI3L1, pro-inflammatory cytokines, and markers of microbial translocation were measured by enzyme-linked immunosorbent assays. Longitudinal clinical characteristics (viral load, neutrophil count, CD4+ and CD8+ T-lymphocyte counts, and antiretroviral (ARV) regimen) were abstracted from patient medical records. One-hundred-and-five (105) CLWH (median age 13 years, 62% female) were included in the study. Seventy-seven (81%) had viral suppression on combination antiviral therapy (cART). The median CHI3L1 level was 25 μg/L (IQR 19–39). CHI3L1 was directly correlated with neutrophil count (ρ = 0.22, *p* = 0.023) and inversely correlated with CD4/CD8 lymphocyte ratio (ρ = −0.35, *p* = 0.00040). Children with detectable viral load had higher levels of CHI3L1 (40 μg/L (interquartile range, IQR 33–44) versus 24 μg/L (IQR 19–35), *p* = 0.0047). CHI3L1 levels were also correlated with markers of microbial translocation soluble CD14 (ρ = 0.26, *p* = 0.010) and lipopolysaccharide-binding protein (ρ = 0.23, *p* = 0.023). We did not detect differences in CHI3L1 between different cART regimens. High levels of neutrophil activation marker CHI3L1 are associated with poor virologic control, immune dysregulation, and microbial translocation in CLWH on cART.

## 1. Introduction

While effective combination antiretroviral therapy (cART) has reduced HIV mortality due to opportunistic infections, disorders associated with chronic inflammation and premature aging are now the leading causes of death in people living with HIV [1]. Neutrophils are important propagators of inflammation during HIV infection [2]. Activated neutrophils contribute to a pro-inflammatory state by promoting macrophage activation, as well as releasing pro-inflammatory cytokines/chemokines, reactive oxygen species (ROS), and neutrophil extracellular traps that cause tissue damage and apoptosis [2]. Several studies have reported increased survival of neutrophils and ROS release in HIV patients on cART [3,4]. Neutrophils have been implicated in the pathogenesis of HIV, as activated neutrophils bound to HIV particles accelerate infection of lymphocytes [5]. Neutrophil activation is also associated with microbial translocation from the gut lumen to the systemic circulation [6], which promotes inflammation and is associated with poor virologic control [7,8,9,10]. As such, activated neutrophils in HIV patients on cART may contribute to a pro-inflammatory state [3,4,5].

Chitinase-3-like protein 1 (CHI3L1), also known as YKL-40, is a secreted glycoprotein produced primarily by activated neutrophils, but also by other cells participating in chronic inflammation: macrophages, fibroblast-like cells, T-lymphocytes, and endothelial cells [11,12,13]. Overexpression of CHI3L1 is characteristic of various inflammatory conditions, and correlates with the severity of coronary and carotid atherosclerosis, thromboembolism, pulmonary arterial hypertension, atrial fibrosis, and epicardial adipose tissue [11,13]. In neutrophils, CHI3L1 is stored in granules and released upon neutrophil activation [14], after which CHI3L1 binds the interleukin-13 receptor α2 (IL-13Rα2)-IL-13 complex and promotes inflammation through apoptosis, pyroptosis, and inflammasome activation [15,16]. CHI3L1 mediates fibrosis and extracellular matrix (ECM) remodelling via stromal cell and fibroblast activation, which may contribute to the pathogenesis of inflammatory diseases, such as atherosclerosis [16,17].

The purpose of this study was to investigate the association between CHI3L1 and markers of virologic control (viral load, CD4/CD8 ratio), inflammation (interleukin [IL]-6, tumor necrosis factor [TNF], C-reactive protein [CRP]), and microbial translocation (soluble CD14 [sCD14] and lipopolysaccharide binding protein [LBP]) in CLWH.

## 2. Materials and Methods

### 2.1. Study Design

We conducted a retrospective cross-sectional study evaluating clinical correlates of HIV disease, chronic inflammation, gut translocation, and CHI3L1 serum level among CLWH within the Early Pediatric Initiation Canada Child Cure Cohort (EPIC^4^) Study [18]. All participating institutions provided ethics approval. Adolescents provided their consent to participate where deemed capable of consent. Parents of children provided informed consent, and assent was also sought from children if developmentally appropriate.

### 2.2. Clinical Definitions

Undetectable viral load was defined as HIV RNA measurement below the quantification limit by the clinical virology laboratory at each center (target not detected, <20, or <40 copies/mL) on the day of the study visit. Sustained virologic suppression (SVS) was defined as undetectable viral load for at least 6 months prior to the study visit. The proportion of life with SVS was determined as the sum of all time periods during which SVS was achieved in days (numerator) divided by participant age in days (denominator). Clinical chart records were used to abstract data on current treatment regimen (antiretroviral exposure).

### 2.3. Measurement of Biomarker Levels

Each clinical site collected whole blood in EDTA tubes, which was shipped to a central laboratory within 24 h for processing. Whole blood samples were centrifuged to separate plasma from cell pellets, and plasma samples were stored at −80 °C. Biomarker levels were quantified using commercially available ELISA kits, according to manufacturer’s instructions (R&D Duoset, Minneapolis, MN, USA). A microplate reader was used to measure optical density; background signal was measured from blank wells on each plate, then subtracted from all samples prior to analysis. A standard curve was subsequently used to determine biomarker concentrations from optical density measurements. All laboratory assays were performed blinded to clinical data.

### 2.4. Statistical Analysis

Since the distributions of biomarker levels were not Gaussian, non-parametric statistical methods were used. Descriptive statistics used the median and interquartile range (IQR) for continuous variables. The median CHI3L1 of the cohort was used to define patients with low CHI3L1 and high CHI3L1 (Table 1). This method was used in previous studies, when the normal range of a biomarker concentration is not well defined [19,20]. Correlations were assessed using Spearman’s rank correlation coefficient (ρ). The independent predictive value of CHI3L1 for the CD4/CD8 ratio and for markers of microbial translocation (sCD14 and LBP) was tested using multivariable linear regression models. Log-transformation of CHI3L1, CD4/CD8 ratio, sCD14, and LBP was used to more closely approximate a normal distribution of the data. Age, sex, and the neutrophil count were included as covariates in the models. The adjusted association between CHI3L1 and the dependent variables was expressed using the regression model coefficient (β) and its 95% confidence interval. Analyses were performed using GraphPad Prism version 6 (GraphPad Software Inc., La Jolla, CA, USA), and R (R Core Team, version 3.3.1).

## 3. Results

A total of 105 CLWH were included, with study visits between February 2015 and December 2016. Characteristics of the cohort are shown in Table 1.

The median plasma concentration of CHI3L1 was 25 μg/L (IQR 19 to 39). Among children with high CHI3L1 (≥sample median), the proportion with undetectable viral load was lower, SVS was lower, the neutrophil count was higher, the frequency of CD4+ T lymphocytes was lower, and the frequency of CD8+ T lymphocytes was higher (Table 1). We also observed a statistically significant correlation between the neutrophil count and CHI3L1 levels (ρ = 0.22, *p* = 0.023, Figure 1). Of note, there was a non-statistically significant difference in the CHI3L1 levels according to ethnicity. Among patients of African, Caribbean, or Black (ACB) ethnicity, the CHI3L1 was 24 μg/L (IQR 18 to 34) compared to 29 μg/L (IQR 20 to 47) among other ethnicities (*p* = 0.13). The neutrophil count was 1.9 × 10^9^/L (IQR 1.3–2.7) among ACB patients and 2.4 × 10^9^/L (2.0–3.5) among other ethnicities (*p* = 0.0018).

We observed higher levels of CHI3L1 among patients with detectable viral load (Figure 2A). Furthermore, CHI3L1 was inversely correlated with the CD4/CD8 ratio, a marker of immune reconstitution following effective cART [21] (ρ = −0.35, *p* = 0.00040, Figure 2B). In a multivariable linear regression model adjusting for the effects of age, sex, and neutrophil count, CHI3L1 remained a statistically significant predictor of the CD4/CD8 ratio (β= −0.29, 95%CI −0.44 to −0.072, *p* = 0.0021). However, there was no statistically significant difference in CHI3L1 levels between children with and without SVS (*p* = 0.12). There was no statistically significant correlation between CHI3L1 levels and the proportion of life with SVS (*p* = 0.59).

We next examined whether CHI3L1 levels were associated with other markers of systemic inflammation. We did not observe statistically significant associations between CHI3L1 levels and those of inflammatory cytokines IL-6 (ρ = −0.020, *p* = 0.84) and TNF (ρ = −0.15, *p* = 0.13), or acute phase reactant CRP (ρ = 0.12, *p* = 0.24). Unlike CHI3L1, IL-6, TNF and CRP levels were similar in patients with detectable versus undetectable VL (*p* > 0.05 for all differences between groups). IL-6 and CRP levels were not correlated with the CD4/CD8 ratio (*p* > 0.05 for both correlation coefficients).

We observed a statistically significant correlation between levels of CHI3L1 and expression of sCD14 (ρ = 0.26, *p* = 0.010, Figure 3A), as well as LBP (ρ = 0.23, *p* = 0.023, Figure 3B). In multivariable linear regression models adjusting for the effects of age, sex, and neutrophil count, CHI3L1 remained a statistically significant predictor of sCD14 levels (β = 0.24, 95%CI 0.069–0.40, *p* = 0.0060) and LBP levels (β = 0.15, 95%CI 0.0046–0.28, *p* = 0.043).

We examined whether CHI3L1 or neutrophil count was associated with cART regimen. We found no associations between the NRTI class or the cART core agent and the neutrophil count or CHI3L1 (Appendix A).

## 4. Discussion

In this study, we found that CHI3L1 levels were associated with higher neutrophil count, poor virologic control (detectable viral load, low CD4/CD8 ratio), and markers of microbial translocation (sCD14 and LBP) in CLWH (Figure 4). Although statistically significant, correlations with CHI3L1 were weak or moderate (ρ between 0.2 and 0.4), suggesting that other factors besides CHI3L1 are involved in virologic control and microbial translocation. The role of CHI3L1 was studied in several chronic inflammatory diseases and cancers; our study extends these findings, demonstrating an association between CHI3L1 and poor virologic control in pediatric HIV infection. Its potential role in the pathogenesis of chronic inflammation in HIV requires further study.

The median CHI3L1 level among CLWH in our study was 25 μg/L (IQR 19 to 39). This is similar to healthy adult controls, based on a study using the same commercial ELISA assay (median 36 μg/L) [22]. On the other hand, CHI3L1 levels in CLWH were lower than in patients with severe acute infections including sepsis (median > 1000 μg/L) [23], SARS-CoV-2 (361 μg/L) [24], and severe malaria (200 μg/L) [25]. Levels were also lower compared to adults with hepatitis B and chronic liver fibrosis (median 460.8 μg/L) [26]. This suggests that the level of CHI3L1 among outpatients with chronic HIV in our study reflects a low level of systemic inflammation, relative to patients hospitalized with acute life-threatening infections or another chronic viral infection (hepatitis B).

CHI3L1 levels were higher in patients with a detectable viral load and in those with a low CD4/CD8 ratio. Incomplete suppression of viral replication may contribute to systemic inflammation, which may explain the association of CHI3L1 with VL. The CD4/CD8 ratio is frequently used as a marker of immune reconstitution after cART initiation in HIV [21,27,28]. We found that high CHI3L1 levels were associated with detectable viral load and lower CD4/CD8 ratio (Figure 2). Of note, a persistently low CD4/CD8 ratio is also independently associated with precursor conditions associated with cardiovascular disease (CVD), such as carotid intima-media thickness and arterial stiffness [29,30]. Given its role in inflammation and its association with low CD4/CD8 ratio, it is tempting to speculate that CHI3L1 may also be a clinically informative marker for CVD risk in CLWH. On the other hand, CHI3L1 levels were not correlated with longer-term and lifelong viral suppression (SVS and proportion of life under SVS). Therefore, CHI3L1 may reflect the acute inflammatory state, rather than chronic or cumulative inflammatory injury.

Lower CHI3L1 levels were associated with African, Caribbean or Black (ACB) ethnicity, although this difference was not statistically significant (Table 1). Lower neutrophil counts in ACB patients (1.9 × 10^9^/L (IQR 1.3–2.7) versus 2.4 × 10^9^/L (IQR 2.0–3.5), *p* = 0.0018) may explain this finding. Of note, benign ethnic neutropenia, due to the Duffy null [Fy(a-b-)] phenotype, is common among individuals of sub-Saharan African ancestry and is not associated with an increased risk of infection [31].

CHI3L1 is a member of the glycoside hydrolase family 18—a group of enzymes that hydrolyze glycosidic bonds in amino polysaccharides [11]. Within this family, CHI3L1 belongs to a group of non-enzymatic chitinase-like proteins that bind chitin, but unlike chitinases, lack enzymatic activity [12,13]. CHI3L1 is expressed primarily by neutrophils, and is stored in intracellular granules before being released upon neutrophil activation during an innate immune response [12,14]. Secreted CHI3L1 binds the interleukin-13 receptor α2 (IL-13Rα2)-IL-13 complex, activating the MAPK/Erk, Akt, and Wnt/β-catenin signaling pathways to trigger a pro-inflammatory response [15,16]. Our study found a direct correlation between CHI3L1 levels and neutrophil count, consistent with a neutrophil source of CHI3L1 (Figure 1). Although antiretroviral medications may affect inflammatory markers [32,33] or neutrophils [4], we did not observe an effect of cART regimen on CHI3L1 levels or neutrophil counts (Appendix A). Our cross-sectional study was not designed to examine CHI3L1 regulation and signaling; however, our observation of elevated CHI3L1 in CLWH with poor virologic control is consistent with its known pro-inflammatory mechanism of action.

Microbial translocation from the gut lumen to the circulation is a driver of systemic inflammation in HIV patients [9]. Due to inflammatory gut dysbiosis in HIV, intestinal barrier function may be compromised, allowing microbial products such as lipopolysaccharide (LPS) to enter the systemic circulation [7,8,9,10]. LPS binds to Toll-like receptor 4 (TLR4), as well as co-receptors CD14 and LBP, to promote monocyte/macrophage-mediated inflammation [7,8,9,10]. We found that CHI3L1 levels were positively correlated with levels of sCD14 and LBP, two markers of microbial translocation (Figure 3). Circulating sCD14 and LBP are also associated with morbidity/mortality, CD8 cell activation, and impaired CD4 cell recovery in patients on cART, suggesting that microbial translocation may alter the immune profile and contribute to poor virologic control in HIV [9,10]. Recent reports suggest that activation of neutrophils in chronic HIV contributes to mucosal inflammation and increased microbial translocation, despite cART [6]. We speculate that CHI3L1 may participate in the cycle of neutrophil activation and microbial translocation in chronic HIV.

We did not find an association between CHI3L1 and levels of classic inflammatory cytokines IL-6, TNF, or the acute phase reactant CRP. Although CHI3L1 secretion by neutrophils can be induced by IL-6 and TNF, it is also induced locally through other pathways by macrophages and neutrophils in infected tissue [11,34]. Therefore, CHI3L1 may represent an independent pro-inflammatory pathway and may be an informative index of “silent” inflammation that is not detected by more commonly used measures of inflammation such as IL-6, TNF, or CRP. Identifying pathways of inflammation that are not currently measured in clinical practice, or that elude standard cytokine panels, may improve prediction and monitoring of chronic inflammation in HIV. A proposed conceptual framework, summarizing the observed associations between CHI3L1 and virologic control, microbial translocation, and systemic inflammation in CLWH is shown in Figure 4.

Our study has several limitations. The cross-sectional study design limits our ability to make causal inferences between CHI3L1, markers of virologic control, and inflammation. Therefore, a prospective cohort study following these markers in CLWH should be considered to assess the relation between CHI3L1 and lifelong virologic control. Long-term prospective studies are similarly required to investigate the clinical significance of CHI3L1 (e.g., if it predicts later complications of systemic inflammation such as CVD). CHI3L1 was measured at a single time point, which could lead to error in case of intercurrent illness at the time of sampling. It would be desirable to have multiple measurements (longitudinal study) to address this limitation. It would also be desirable to have a control group of healthy children without HIV infection to compare CHI3L1 levels in normal individuals to CLWH in our cohort. The number of patients on integrase strand transfer inhibitors was small in our study, reflecting a historical cohort with a higher use of protease inhibitors than would be seen in modern practice. Finally, our findings should be extrapolated with caution to treatment centres in low- and middle-income countries, where a large burden of HIV occurs. Further studies in these countries are warranted to assess the global relevance of CHI3L1 in CLWH. Strengths of this study include (a) generalizability of findings to pediatric HIV care centres in high-income countries thanks to the multicentre study design; (b) detailed historical virologic and clinical data for our cohort; and (c) high proportion of CLWH with SVS on cART.

In summary, we have shown that CHI3L1 is associated with poor virologic control, immune dysregulation, and microbial translocation in CLWH. The clinical significance of elevated CHI3L1 levels in CLWH is not known; however, chronic immune activation and inflammation contribute to the pathogenesis of CVD in the context of HIV infection [30,35]. CLWH develop premature atherosclerosis, dyslipidemias, increased arterial stiffness, increased carotid intimal media thickness, coronary arteriopathy, and congestive heart failure [30]. High levels of CHI3L1 may therefore be a poor prognostic marker in people living with HIV, in whom CVD are a leading cause of death [1]. Given the interactions of CHI3L1 and inflammation, we posit that CHI3L1 may be a useful tool to monitor systemic inflammation, with potential predictive value for inflammatory injury or CVD events. Recent studies have suggested that CHI3L1 may be a therapeutic target in a broad range of conditions, including COVID-19, bone metabolism, and breast cancer [36,37,38]. We suggest that antibody-based and small molecule CHI3L1 inhibitors could also be tested as inflammatory modulators in HIV. Future studies are needed to elucidate the mechanism of immune activation involving CHI3L1 in HIV. This may guide further research into improved management of long-term, non-infectious complications of chronic HIV infection.

## Figures and Tables

**Figure 1 viruses-14-02602-f001:**
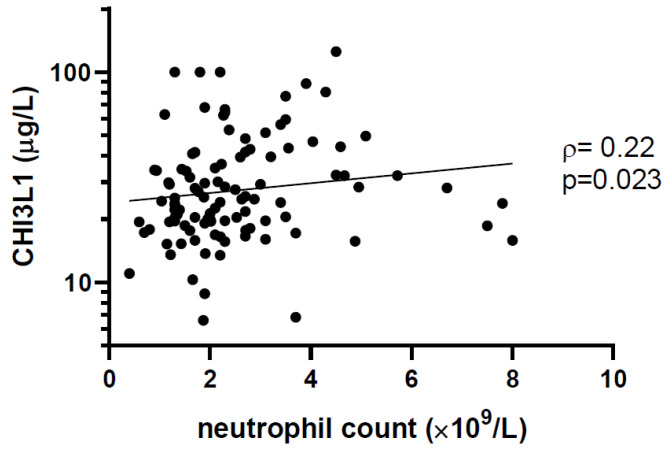
Plasma CHI3L1 level is correlated with neutrophil count. The concentration of CHI3L1 was measured by enzyme-linked immunosorbent assay (ELISA), and neutrophil counts were taken from patient clinical records. Non-parametric Spearman’s rank correlation coefficient (ρ) and the associated *p*-values are indicated.

**Figure 2 viruses-14-02602-f002:**
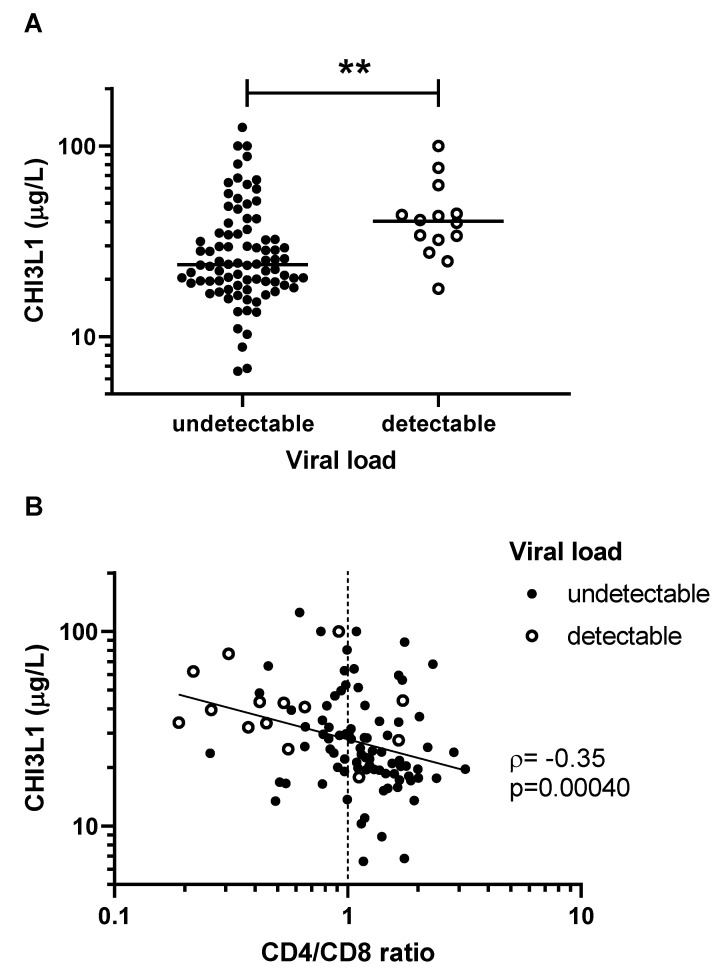
CHI3L1 is associated with poor virologic control among children living with HIV. (**A**) Patients with detectable viral load (solid circles) had significantly more CHI3L1 than those with undetectable viral load (open circles) (** *p* = 0.0047). (**B**) CHI3L1 and CD4/CD8 ratio were inversely correlated. Concentrations of CHI3L1 were measured by ELISA, and CD4/CD8 lymphocyte counts were taken from patient clinical chart records. Non-parametric Spearman’s rank correlation coefficient (ρ) and the associated *p*-values are indicated.

**Figure 3 viruses-14-02602-f003:**
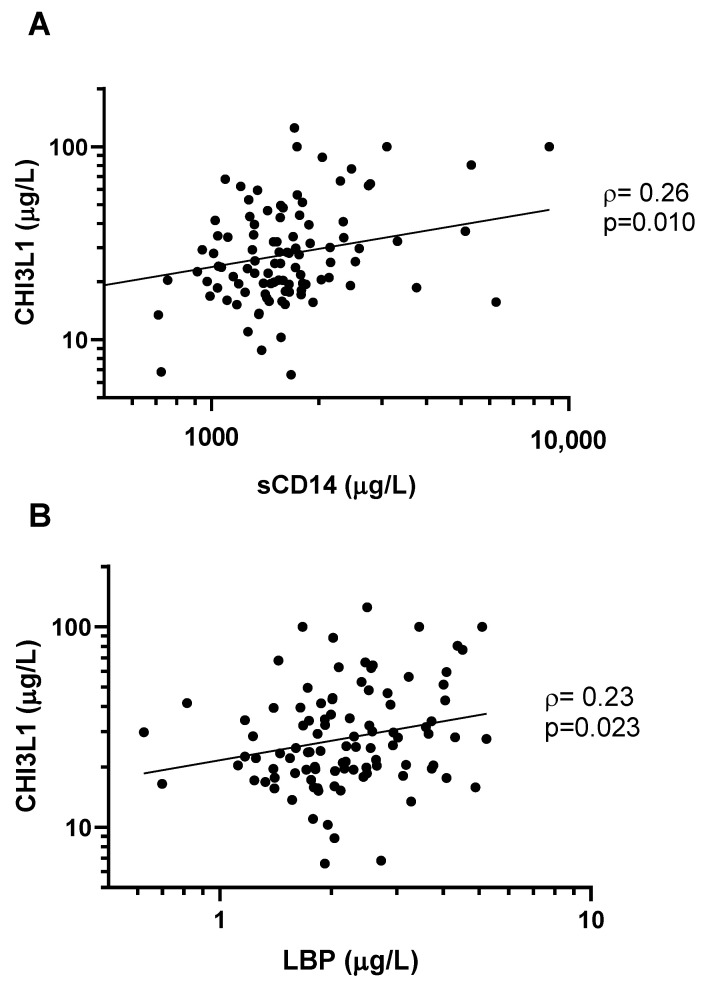
CHI3L1 is correlated with markers of microbial translocation. CHI3L1 was directly correlated with (**A**) sCD14 (ρ = 0.26, *p* = 0.010), and (**B**) LBP (ρ = 0.23, *p* = 0.023). Concentrations of all biomarkers were measured with an ELISA. Non-parametric Spearman’s rank correlation coefficient (ρ) and the associated *p*-values are indicated.

**Figure 4 viruses-14-02602-f004:**
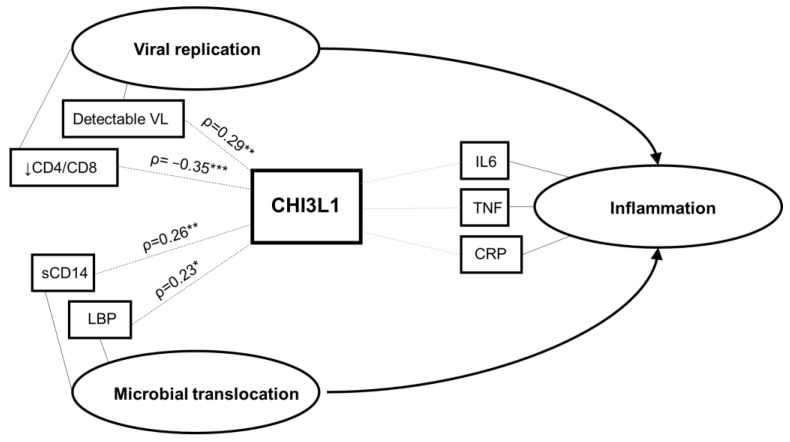
Conceptual framework. Based on the study findings, the hypothesized relationship between CHI3L1, viral replication, microbial translocation, and systemic inflammation is illustrated. Ovals represent latent constructs and rectangles represent measured variables. VL, viral load; IL-6, interleukin-6; TNF, tumor necrosis factor; CRP, C-reactive protein; sCD14, soluble CD14; LPB, lipopolysaccharide binding protein. * *p* < 0.05; ** *p* < 0.01; *** *p* < 0.001.

**Table 1 viruses-14-02602-t001:** Characteristics of 105 children living with HIV, according to plasma chitinase-3-like-1 (CHI3L1) protein level.

Characteristic	Overall Cohort (N = 105)	Low CHI3L1 ^1^ (N = 52)	High CHI3L1 ^1^ (N = 53)	*p*-Value
Demographics				
Age (yr), median (IQR)	13 (8.6–17)	13 (8.5–16)	14 (8.6–18)	0.28
<8 years, *n* (%)	22 (21)	10 (19)	12 (23)	0.70
8 to <12 years, *n* (%)	27 (26)	13 (25)	14 (26)	
12 to <16 years, *n* (%)	20 (19)	9 (17)	11 (21)	
≥16 years, *n* (%)	36 (34)	21 (40)	15 (28)	
Female sex, *n* (%)	62 (59)	33 (63)	29 (55)	0.48
Ethnicity (mother), *n* (%)				0.16
African, Caribbean, or Black	67 (64)	38 (73)	29 (55)	
White	12 (11)	8 (15)	4 (7.5)	
First Nations	7 (6.7)	2 (3.8)	5 (9.4)	
Other	16 (15)	4 (7.7)	12 (23)	
Unknown	3 (2.9)	0 (0)	3 (5.7)	
HIV Clade, *n* (%)				0.21
Clade A	4 (3.8)	0 (0)	4 (7.5)	
Clade B	32 (30)	18 (35)	14 (26)	
Clade C	21 (20)	10 (19)	11 (21)	
Other	22 (21)	11 (21)	11 (21)	
Unknown	26 (25)	13 (25)	13 (25)	
Historical HIV control, median (IQR)				
Age at initiation of any ARVs (yr)	2.1 (0.38–5.5)	3.4 (0.84–5.5)	1.7 (0.36–5.3)	0.32
Undetectable viral load, *n* (%)	84 (86)	46 (96)	38 (76)	0.0026
Proportion SVS, *n* (%)	81 (77)	45 (87)	36 (68)	0.041
Age at SVS (yr) ^2^	6.6 (3–11)	6.6 (3.8–12)	6.6 (2.4–8.9)	0.27
Duration of SVS (yr) ^2^	6.0 (3–8.4)	4.9 (2.4–7.3)	6.1 (3.7–9.8)	0.11
Current cART regimen, *n* (%)				0.096
NNRTI-based	41 (39)	22 (42)	19 (36)	
Protease-inhibitor-based	23 (22)	11 (21)	12 (23)	
Integrase inhibitor-based	17 (16)	12 (23)	5 (9.4)	
No ARVs	10 (9.5)	2 (3.8)	8 (15)	
Other ^3^	14 (13)	5 (9.6)	9 (17)	
Immunologic variables, median (IQR)				
Neutrophil count (×10^9^/L)	2.2 (1.6–3.1)	2.0 (1.3–2.7)	2.3 (1.8–3.5)	0.041
Neutrophil percent	41 (33–52)	38 (32–50)	43 (36–57)	0.11
CD4+ T-cell count (×10^9^/L)	770 (550–1100)	830 (590–1100)	690 (510–990)	0.13
CD4+ T-cell percent	37 (30–42)	40 (35–43)	33 (26–40)	0.0011
CD4 lifetime nadir (×10^9^/L)	440 (280–570)	440 (290–570)	440 (250–610)	0.79
CD4 lifetime nadir percent	20 (14–29)	23 (15–29)	19 (13–26)	0.12
CD8+ T-cell count (×10^9^/L)	740 (520–1000)	620 (460–860)	810 (600–1100)	0.0071
CD8+ T-cell percent	33 (27–38)	30 (25–35)	36 (29–44)	0.00039

SVS, sustained virologic suppression; cART, combination antiretroviral therapy; IQR, interquartile range; NNRTI, non-nucleoside reverse transcriptase inhibitor. ^1^ High CHI3L1 was defined as a plasma concentration ≥ sample median (25 μg/L). ^2^ Among those with VS (*n* = 81). ^3^ Other cART regimens included: lopinavir (LPV) + ritonavir (boost) (RTVb) + raltegravir (RAL) (*n* = 2); atazanavir (ATZ) + RTVb + RAL (*n* = 1); ATZ + elvitegravir/cobicistat (EVG/COBI) (*n* = 1); darunavir (DRV) + EVG/COBI (*n* = 1); etravirine (ETR) + RAL (*n* = 1); RAL + maraviroc (*n* = 1); ETR + DRV + RTV + dolutegravir (DTG) (*n* = 1); ETR + DRV + RTVb + RAL (*n* = 1); and unknown (*n* = 5). Bars with whiskers represent median and interquartile range.

## Data Availability

Data are available upon request from the study authors.

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
