# Peer review of "Chitinase-3-like Protein 1 Is Associated with Poor Virologic Control and Immune Activation in Children Living with HIV"

_viruses, 2022, doi:10.3390/v14122602_

Round 1

Reviewer 1 Report

Thank you for completing this important work. We have wished to better  understand the mechanisms for observed inflammatory states of HIV infected children and adults.  

In the US, we have replaced "vertically infected" with perinatally infected to encompass babies who became HIV infected during pregnancy, childbirth and breastfeeding; unless you specifically can state that the children included were infected in pregnancy or childbirth, would recommend changing the phrase used.  

I found the clinical definitions used in this study to be relevant to clinicians and researchers.  

The methodology makes good sense. I do wish that more longitudinal data or multiple time points could have been used, as a single timepoint could be biased by intercurrent illness. This should be stated in the limitations.

It would be interesting to see subgroups broken down by age groups (for example, 8-12 years of age, 13-16 years of age) as immune phenomena can shift through adolescence. This should also be stated in the limitations.

There were non-statistically significant differences in the ethnicities in the low and high SH13LI groups, this should be mentioned. Also, attention should be paid to racial and ethnic differences in neutrophil counts. 

It would be interesting to note if any of the subjects with an elevated CD4:8 ratio had recently started cART or if they had experienced IRIS. 

Please clarify the text to state that you looked for correlation between ART classes rather than individual drugs. 

Author Response

We thank the reviewer for this valuable feedback. We have responded in a point-by-point manner to each of the comments, below. A revised manuscript with tracked changes, as well as a clean version, are attached.

Thank you for completing this important work. We have wished to better  understand the mechanisms for observed inflammatory states of HIV infected children and adults.  

We thank the reviewer for this remark.

In the US, we have replaced "vertically infected" with perinatally infected to encompass babies who became HIV infected during pregnancy, childbirth and breastfeeding; unless you specifically can state that the children included were infected in pregnancy or childbirth, would recommend changing the phrase used.  

We thank the reviewer for this recommendation. We have changed the wording from “vertically infected” to “perinatally infected” (Page 1)

I found the clinical definitions used in this study to be relevant to clinicians and researchers.  

Thank you for this remark.

The methodology makes good sense. I do wish that more longitudinal data or multiple time points could have been used, as a single timepoint could be biased by intercurrent illness. This should be stated in the limitations.

We agree with the reviewer that longitudinal data (multiple time points) would be helpful. We have added this as a limitation in the Discussion of the revised manuscript, as follows:

“CHI3L1 was measured at a single time point, which could lead to error in case of intercurrent illness at the time of sampling. It would be desirable to have multiple measurements (longitudinal study) to address this limitation.”

It would be interesting to see subgroups broken down by age groups (for example, 8-12 years of age, 13-16 years of age) as immune phenomena can shift through adolescence. This should also be stated in the limitations.

We thank the reviewer for this suggestion. In the revised manuscript, we have added the following age categories to Table 1: <8 years, 8 to <12 years, 12 to <16 years, and ≥16 years. We did not find a statistically significant difference in the CHI3L1 levels between different age categories.

There were non-statistically significant differences in the ethnicities in the low and high SH13LI groups, this should be mentioned. Also, attention should be paid to racial and ethnic differences in neutrophil counts. 

Thank you for this astute remark. As noted by the reviewer, patients of African, Caribbean or Black (ACB) ethnicity appear to have lower CHI3L1 levels than other ethnic groups, although the difference was not statistically significant (Table 1). Among ACB patient, the CHI3L1 was 24 μg/L (IQR 18 to 34) compared to 29 μg/L (IQR 20 to 47) among other ethnicities (p=0.13). The neutrophil count was 1.9 ×109/L (IQR 1.3-2.7) among ACB patients and 2.4 ×109/L (2.0-3.5) among other ethnicities (p=0.0018). This difference may be explained by benign ethnic neutropenia (BEN). BEN is associated with sub-Saharan African ancestry, and is not associated with an increased risk of infection. The mechanism of lower ANC levels may be explained by the Duffy null [Fy(a-b-)] phenotype, which is protective against malaria, and may be selected in populations with high exposure to malaria. We have included this interesting observation in the Results and Discussion sections of the revised manuscript.

It would be interesting to note if any of the subjects with an elevated CD4:8 ratio had recently started cART or if they had experienced IRIS. 

Thank you for this query. Although we did not specifically collect data on IRIS in our cohort, bloodwork was deferred in patients with fever or intercurrent illness; therefore, it is unlikely that patients in our cohort had IRIS.

To examine whether recent initiation of cART was associated with elevated CD4:CD8 ratio, we began by examining the correlation between the time since treatment initiation and the CD4:CD8 ratio. We did not find a statistically significant correlation (ρ=0.0030, p=0.9763). Patients who had started cART within the past year had a lower CD4:CD8 ratio (0.65, IQR 0.56-0.79) than those who had initiated cART more than one year prior (1.1 IQR 0.87-1.6) (p=0.0089). This is consistent with gradual immune reconstitution with rise in CD4 counts following sustained viral suppression.

Please clarify the text to state that you looked for correlation between ART classes rather than individual drugs

Thank you for this comment. We have changed the sentence in the Results section of the manuscript as follows:

We examined whether CHI3L1 or neutrophil count was associated with cART regimen. We found no associations between the NRTI class or the cART core agent and the neutrophil count or CHI3L1 (Supplemental Figure S1).

Reviewer 2 Report

The article by Bernard and collaborators identifies chitinase-3-like protein 1 as associated with poor HIV control and indicators of microbial translocation. This is an interesting observation, but there are several limitations of the study. 

·      The first is the lack of comparisons to healthy controls. Without these, the relationship between CHI3L1 and HIV is tangential. Are CHI3L1 levels different when comparing CLWH to age-matched healthy children? Although unlikely, it is possible that the associations, or some of the associations, are independent of HIV and related to immunologic variables shown in Table 1. 

·      Another concern is that the goals of the study are unclear. The stated purpose is to investigate an association. Later in the paper the authors propose it may be a marker of CVD risk, but is this relevant in the pediatric population? How is it more informative than other cytokines that have been associated with HIV infection (see PMID28053103 for example)? Is it a therapeutic target? 

·      Some of the conclusions are overstated. Yes, the correlations are significant, but the rho values suggest they are very week. Spearman correlations are often significant when a high number of samples are tested. A statistician should be consulted.

·      If the goal is to use CHI3L1 as a predictor or biomarker an interaction analysis should be done to see if the association with control/translocation is stronger when considered in a multivariable analysis with the immunologic variables. A statistician could be consulted for this.

Author Response

We thank the reviewer for this valuable feedback. We have responded in a point-by-point manner to each of the comments, below. A revised manuscript with tracked changes, as well as a clean version, are attached.

The article by Bernard and collaborators identifies chitinase-3-like protein 1 as associated with poor HIV control and indicators of microbial translocation. This is an interesting observation, but there are several limitations of the study. 

We thank the reviewer for this remark and we trust that we have addressed the limitations below.

  • The first is the lack of comparisons to healthy controls. Without these, the relationship between CHI3L1 and HIV is tangential. Are CHI3L1 levels different when comparing CLWH to age-matched healthy children? Although unlikely, it is possible that the associations, or some of the associations, are independent of HIV and related to immunologic variables shown in Table 1. 

We agree with the reviewer that healthy controls would provide an interesting control group. Unfortunately, the EPIC4 study enrolled children living with HIV (CLWH) from eight pediatric HIV care centres across Canada, but did not enrol healthy controls.

The difference in CHI3L1 levels between patients with detectable viral load and those CLWH with suppressed viral load suggests a direct association between CHI3L1 and HIV replication. We agree with the reviewer that the observed associations between CD4/CD8ratio, microbial translocation and CHI3L1 could be related to immunologic variables (e.g., neutrophil count). However, derangements in these immunologic variables are likely related to the chronic HIV infection.

In response to this comment, we have included in the limitations section of the Discussion: “It would also be desirable to have a control group of healthy children without HIV infection to compare CHI3L1 levels in normal individuals to CLWH in our cohort.”

  • Another concern is that the goals of the study are unclear. The stated purpose is to investigate an association. Later in the paper the authors propose it may be a marker of CVD risk, but is this relevant in the pediatric population? How is it more informative than other cytokines that have been associated with HIV infection (see PMID28053103 for example)? Is it a therapeutic target? 

We thank the reviewer for this comment. The purpose of our study was to investigate the association between CHI3L1 and markers of virologic control, inflammation, and microbial translocation in CLWH. We stated this objective in the final paragraph of the Introduction (Page 1).

Later, we discuss the possible implication of these findings (Page 11). We note that the clinical significance of elevated CHI3L1 levels in CLWH is not known, but that chronic immune activation and inflammation contribute to the pathogenesis of CVD in the context of HIV infection. This prompted us to speculate about the possible role of CHI3L1 as a CVD risk marker.

This is relevant to children because children living with HIV face a lifetime of chronic inflammation. They develop premature atherosclerosis, dyslipidemias, increased arterial stiffness, increased carotid intimal media thickness, coronary arteriopathy, and congestive heart failure. High levels of CHI3L1 may therefore be a poor prognostic marker for future CVD, a leading cause of death in adulthood (i.e., developmental origin of adult pathology).

CHI3L1 and other markers of inflammation (e.g., CCL14, CCL21, CCL27, and XCL1 measured in Jacobs et al. Cytokines Elevated in HIV Elite Controllers Reduce HIV Replication In Vitro and Modulate HIV Restriction Factor Expression. J Virol. 2017 Feb 28;91(6):e02051-16) have not been evaluated in prospective studies for their predictive value for clinical or surrogate clinical endpoints.

The reviewer makes an intriguing point, wondering whether CHI3L1 could be a therapeutic target. A recent paper demonstrated the efficacy of blocking CHI3L1 in COVID-19 models in vitro and in vivo using antibody-based and small molecule CHI3L1 inhibitors (Kamle et al. Chitinase 3-like-1 is a therapeutic target that mediates the effects of aging in COVID-19. JCI Insight. 2021 Nov 8; 6(21): e148749.) Another recent article described targeting CHI3L1 in a non-infectious disease, osteoporosis (Park et al., Chi3L1 is a therapeutic target in bone metabolism and a potential clinical marker in patients with osteoporosis. Pharmacol Res. 2022 Oct;184:106423). Taken together, these studies suggest that CHI3L1 may indeed be a therapeutic target for inhibition of chronic inflammation in HIV. This hypothesis warrants further study.

In response to this comment, we have added a sentence to the Discussion, speculating on CHI3L1 as a potential therapeutic target to block inflammation in HIV:

“Recent studies have suggested that CHI3L1 may be a therapeutic target in COVID-19, bone metabolism, and breast cancer. We suggest that antibody-based and small molecule CHI3L1 inhibitors could also be tested as inflammatory modulators in HIV.”

  • Some of the conclusions are overstated. Yes, the correlations are significant, but the rho values suggest they are very week. Spearman correlations are often significant when a high number of samples are tested. A statistician should be consulted.

Thank you for this comment. We have restated the conclusions to be less definitive and highlight that the rank correlation coefficients indicate a weak or moderate correlation between CHI3L1 and markers of virologic control and inflammation.

We have removed the following sentence (Page 11): “These findings suggest that CHI3L1 may be a clinically informative biomarker of systemic inflammation in CLWH.”

We have added the following sentence (Page 11): “Although statistically significant, correlations with CHI3L1 were weak or moderate (ρ between 0.2 and 0.4), suggesting that other factors besides CHI3L1 are involved in virologic control and microbial translocation.”

  • If the goal is to use CHI3L1 as a predictor or biomarker an interaction analysis should be done to see if the association with control/translocation is stronger when considered in a multivariable analysis with the immunologic variables. A statistician could be consulted for this.

We thank the reviewer for this suggestion, which strengthens our findings.

Having observed a significant correlation between CHI3L1 and the CD4/CD8 ratio (Figure 2B), we examined the association in a multivariable linear regression model with the CD4/CD8 ratio as the dependent variable, and CHI3L1, age, sex, and neutrophil count as independent predictors. Because of the non-Gaussian distribution of the CD4/CD8 ratio and the CHI3L1 levels, we used log-transformation of the data to better approximate a normal distribution. The association between CHI3L1 and the CD4/CD8 ratio remained statistically significant, after adjustment for age, sex, and neutrophil count as covariates (p=0.0021).

Similarly, we observed a significant correlation between CHI3L1 and sCD14 (Figure 3A) as well as between CHI3L1 and LBP (Figure 3B). We next examined the associations in multivariable linear regression models with sCD14 or LBP as dependent variables, and CHI3L1, age, sex, and neutrophil count as independent predictors. The association between CHI3L1 and sCD14 remained statistically significant after adjustment for the age, sex, and neutrophil count as covariates (p=0.0060). The association between CHI3L1 and LBP also remained statistically significant (p=0.040).

To address the reviewer’s suggestion, we have added the following sentences to the Methods (Page 3) and the Results (Pages 6 and 7):

Methods (Page 3): “The independent predictive value of CHI3L1 for the CD4/CD8 ratio and for markers of microbial translocation (sCD14 and LBP) was tested using multivariable linear regression models. Log-transformation of CHI3L1, CD4/CD8 ratio, sCD14, and LBP was used to more closely approximate a normal distribution of the data. Age, sex, and the neutrophil count were included as covariates in the models. The adjusted association between CHI3L1 and the dependent variables was expressed using the regression model coefficient (β) and its 95% confidence interval.”

Results (Page 6): “In a multivariable linear regression model adjusting for the effects of age, sex, and neutrophil count, CHI3L1 remained a statistically significant predictor of the CD4/CD8 ratio (β= -0.29, 95%CI -0.44 to -0.072, p=0.0021).”

Results (Page 7): “In multivariable linear regression models adjusting for the effects of age, sex, and neutrophil count, CHI3L1 remained a statistically significant predictor of sCD14 levels (β=0.24, 95%CI 0.069-0.40, p=0.0060) and LBP levels (β=0.15, 95%CI 0.0046-0.28, p=0.043).”